# How are oxygen budgets influenced by dissolved iron and growth of oxygenic phototrophs in an iron-rich spring system? Initial results from the Espan Spring in Fürth, Germany

Inga Köhler[1], Raul E. Martinez[2], David Piatka.[1], Achim J. Herrmann[3], Arianna Gallo[3], Michelle M. Gehringer[3], Johannes A.C. Barth[1]

[1]Department of Geography and Geosiences, GeoZentrum Nordbayern, Schlossgarten 5, Friedrich-Alexander-Universität FAU, Erlangen, 91054, Germany

[2]Max-Planck-Institute for Biogeochemistry, Jena, 07745, Germany

[3]Division of Microbiology, Technische Universität, Kaiserslautern, 67663 Germany

*Correspondence to*: Inga Köhler (inga_koehler@gmx.de)

**Abstract.** At present most knowledge on the impact of iron on $^{18}O/^{16}O$ ratios (i.e. $\delta^{18}O$) of dissolved oxygen (DO) under circum-neutral conditions stems from experiments carried out under controlled laboratory conditions. These showed that iron oxidation leads to an increase in $\delta^{18}O_{DO}$ values**.** Here we present the first study on effects of elevated Fe(II) concentrations on the $\delta^{18}O_{DO}$ in a natural, iron-rich circum-neutral watercourse. Our results show that iron oxidation was the major factor to cause rising oxygen isotopes in the first 85 meters of the system in the cold season (February) and for the first 15 meters during the warm season (May). Further along the course of the stream, the $\delta^{18}O_{DO}$ decreased towards values known for atmospheric equilibration at 24.6 ‰ during both seasons. Possible drivers for this decrease may be reduced iron oxidation, increased atmospheric exchange and DO production by oxygenic phototrophic algae mats. In the cold season, the $\delta^{18}O_{DO}$ values stabilized around atmospheric equilibrium, whereas in the warm season stronger influences by oxygenic photosynthesis caused values down to +21.8 ‰. In the warm season after 145 meters downstream of the spring, the $\delta^{18}O_{DO}$ increased again until it reached atmospheric equilibrium. This trend can be explained by a respiratory consumption of DO combined with a relative decrease in photosynthetic activity and increasing atmospheric influences. Our study shows that dissolved Fe(II) can exert strong effects on the $\delta^{18}O_{DO}$ of a natural circum-neutral spring system even under constant supply of atmospheric $O_2$. However, in the presence of active photosynthesis, with active supply of $O_2$ to the system, direct effects of Fe oxidation on the $\delta^{18}O_{DO}$ value becomes masked. Nonetheless, critical Fe(II) concentrations may indirectly control DO budgets by enhancing photosynthesis, particularly if cyanobacteria are involved.

## 1 Introduction

Oxygen is the most abundant (45.2 %) and iron the fourth most abundant (5.8 %) element on earth (Skinner, 1979). Such huge global reservoirs render these elements critically important in global biogeochemical cycles. In addition, their reactivity is exceptional: $O_2$ is a powerful oxidation agent while Fe can cover oxidation states from –4 to +7 in extreme cases, with the most commonly known ones being 0, +2 and +3 (Lu et al., 2016).

Iron is also an essential trace element in many biological processes, including photosynthesis, oxygen transport and DNA biosynthesis (Kappler et al., 2021). This closely links to the formation and dissolution of Fe oxides. These common forms of metal oxides may enhance or reduce availabilities of both elements in the water column and pore waters and thus may largely regulate aqueous life.

In aqueous environments, dissolved oxygen (DO) is one of the most essential ecosystem parameters and, despite its moderate solubility (e.g. 9.3 mg/L at 20 °C), it assumes a central role in respiration, primary production and Fe-oxidation (Pusch, 1996). The concentration of DO coupled to its stable isotope $^{18}O/^{16}O$ ratios (i.e. $\delta^{18}O$) can

yield additional information about sources and sinks, including atmospheric input, photosynthesis, respiration and mineral oxidation.

When equilibrated with the atmosphere, $\delta^{18}O_{DO}$ values typically range around a value of + 24.6 ‰ (Mader et al., 2017) while photosynthesis and respiration can change these isotope ratios (Guy et al., 1993; Kroopnick, 1975). The splitting of water molecules during photosynthesis hardly produces an isotope discrimination and the resulting DO should have the same isotope value as the surrounding water (Guy et. al., 1993; Eisenstadt et al., 2010). Meteoric water in temperate climates is normally depleted in $^{18}O$ and therefore the photosynthetic oxygen in these areas varies between - 10 to – 5 ‰ (Quay et. al., 1995; Wang and Veizer, 2000). Respiration, on the other hand, preferentially accumulates $^{16}O$ and enriches the remaining DO in $^{18}O$. This process yields $\delta^{18}O_{DO}$ values between + 24.6 and + 40 ‰ (Guy et. al., 1993).

Additionally, oxidation of metals such as Fe also lead to increases in $\delta^{18}O_{DO}$ (Lloyd, 1968; Taylor and Wheeler, 1984; Wassenaar and Hendry, 2007; Oba and Poulsen, 2009 a,b; Pati, 2016). Mostly, the impacts of Fe oxidation on $\delta^{18}O_{DO}$ values have been investigated experimentally under controlled conditions (Oba and Poulson, 2009b; Pati et al., 2016). As a new aspect, these dynamics were not studied in open water systems such as springs and rivers so far. New field investigations might reconcile variations in the fractionation factors obtained in the abovementioned studies. At current they are thought to result from differences in temperature, pH and initial Fe(II) concentrations that could be outlined under abiotic conditions.

Dissolved Fe(II) in natural systems may have primary and secondary impacts on DO concentration and its $\delta^{18}O_{DO}$ values. The primary influence originates from the $O_2$ binding by iron oxidation (equation 1). This leads to decreases of the DO and causes simultaneous increases of $\delta^{18}O_{DO}$ values (Wassenaar and Hendry, 2007; Smith et al., 2011; Parker et al., 2012 and Gammons et al., 2014).

(1)   $4\ Fe^{2+} + O_2 + 4\ H^+ \rightarrow 4\ Fe^{3+} + 2\ H_2O$

Dissolved Fe(II) can also have secondary (i.e. indirect) influences on the DO content and the $\delta^{18}O_{DO}$. This happens when it acts as an essential micronutrient to cause growth-stimulating effects on $O_2$-producing and respiring microorganisms. These influences of Fe(II) on DO and $\delta^{18}O_{DO}$ in circum-neutral aquatic systems have so far received little attention because of the following reasons:

(1) Fe oxidation often masks $\delta^{18}O_{DO}$ values created by respiration, photosynthetic and atmospheric oxygen and

(2) adequate Fe(II)-rich circum-neutral model systems are scarce on modern earth. This is due to the high reactivity of iron with DO.

To the best of our knowledge, no study so far has systematically investigated the influences of elevated Fe(II) concentrations on $\delta^{18}O_{DO}$ values in a natural and circum-neutral iron-rich system. In order to bridge this gap, we investigated the aqueous chemistry and $\delta^{18}O_{DO}$ values in the iron-rich Espan Spring in Fürth, Germany (Fig. 1). This Fe(II)-rich artesian spring offers a complex biogeochemical natural field site to analyse effects of different Fe(II) contents on the DO and $\delta^{18}O_{DO}$ values.

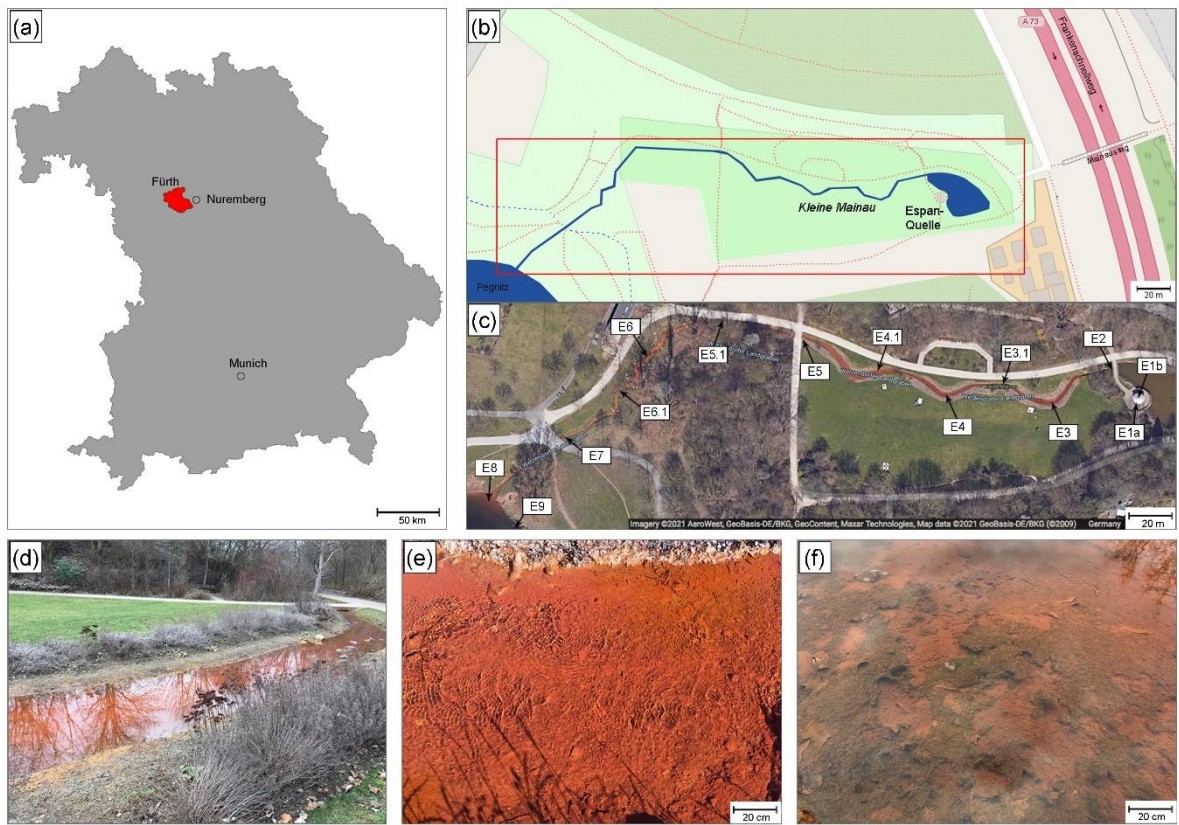

**Figure 1** Overview of the Espan Spring in Fürth, Germany. a) and b): Location of the spring in Bavaria and the city of Fürth. c) Satellite image of the spring by Google maps showing the distinct red colour. d) to f) Detailed photos of the system. d) displays the stream between sampling points E4 and E5, e) shows sampling point E3 (with the bank of the water line in the upper part of the picture) and f) displays sampling point E4.1 with algae and cyanobacteria mats.

The aims of this study were to establish an inventory of biology together with Fe and oxygen budgets in this natural spring and stream system. We further aimed to investigate how increased Fe(II)-levels influence the oxygen budget of the system and whether a combination of DO and $\delta^{18}O_{DO}$ measurements can help to assess this effect. This is also timely because environmental impacts of Fe(II) become increasingly recognised for their negative effects on ecosystems such as with the browning or brownification phenomenon (Kritzberg and Ekström, 2011; Weyhenmeyer et al., 2014; Kritzberg et al., 2020). During this process, increased iron levels can consume oxygen, cause algae blooms and reduce water quality and thus may affect aqueous ecosystems and their services. Here we describe a first complete spatial sampling campaign in the cold and warm season with Fe(II), Fe(III), DO and its stable $^{18}O/^{16}O$ isotope ratios together with field parameters (pH, T, DO, pe, electrical conductivity).. This study contributes to the knowledge of Fe oxidation in natural systems and delivers implications of hardly explored seasonal dynamics in Fe(II) rich systems.

## 2 Methods

### 2.1 Study site

The Espan Spring is located in the city of Fürth, Germany (49°28'15.8"N 11°00'53.0"E, Fig. 1). It is an artesian
spring that originates from a confined aquifer that was tapped by a drilling project in 1935 from a depth of 448.5
m below ground. The water originates from the so called "lower mineral water horizon". This horizon is dominated
by artesian inflow from the lower Buntsandstein Formation. The Buntsandstein in Fürth consists of red sandstone
layers that are composed of light reddish to yellowish-white-grey sandstones of different grain sizes. The
sandstones are intercalated with various rubble, conglomerate, and clay layers as well as thin gypsum and salt
(Birzer, 1936). Three noticeable conglomerate layers are present in the sequence of Buntsandstein layers. Birzer
(1936) distinguished the Upper Buntsandstein from the Upper Main Buntsandstein by the Main Conglomerate
which can be found at a depth of 321 to 324 m. The Middle Boulder Layer at a depth of 370 to 371 m separates
the Upper from the Lower Main Buntsandstein and the so-called Eck'sche Conglomerate at a depth of 433 to 440
m which separates the Lower Main Buntsandstein from the Lower Buntsandstein (Birzer, 1936).
At a depth of 370 to 439 m, mineral water flows into the borehole from the Upper and Lower Main Bunter
Sandstone and from the Eck conglomerate. This water, which is caught in the red sandstone and has a temperature
of about +22°C, was called the "Lower Mineral Water Horizon"; in 1936 its yield was about 10 L $s^{-1}$ at a water
temperature of +23°C (Kühnau 1938). The water of this lower spring horizon is under artesian pressure and exits
the spring with a head of 13 m above ground level (Birzer, 1936). Nowadays the Espan Spring has a constant yield
of about 5 L $s^{-1}$.
After the water exits the basin in a pavilion with a temperature of ~ 20 °C, it discharges into a stream of about 300
m length that is known as the "Wetzendorfer Landgraben (WL)". This small stream drains into the Pegnitz River
without any further tributaries (Fig. 1b, c). The water can be classified as a $Na-Ca-Cl-SO_4$ mineral water with
initially undersaturated DO values of 2.3 mg/L and Fe(II) contents of up to 6.6 mg/L (Table 1). Figure 1c shows
an aerial image of the spring and stream system that shows a distinct red coloring of the stream bed. The most
plausible explanation for this coloring are iron-oxide-precipitates (Fig. 1 d, e). The WL has a water depth between
8-10 cm and shows little fluctuations.

**2.2 Sampling procedures**

Two field campaigns were performed in February and May 2020, during which water was collected at 14 locations
along the stream. The onsite parameters pH (±0.05; instrument precision), temperature (±0.1 °C), electrical
conductivity, Eh and DO (all ±2 %) were measured with a HACH HQ 40d multi parameter instrument. Alkalinity
titrations were carried out with a Hach Titrator with a bromocresol-green indicator. Fe(II) and Fe(III) contents were
measured using an iron (II/III) cuvette test set by Hach in combination with a portable Hach spectrophotometer
(model DR 2800).
Samples for $^{18}O/^{16}O$ ratios of DO were collected in 12-mL Exetainers (Labco Ltd. Lampeter, U.K.) that were
prepared with 10 μL of a saturated $HgCl_2$ solution to prevent secondary biological activity after sampling
(Wassenaar and Koehler, 1999; Parker et al., 2005 and 2010). The Exetainers were filled with syringe-filtered
water via 0.45 μm pore size nylon filters until they were entirely full and free of air bubbles. They were then
carefully closed with screw caps with a butyl septum in order to avoid atmospheric contamination. Test series
showed that the amount of atmospheric contamination during this filling procedure is usually negligible (Mader et
al. 2018).
Samples for water isotopes were collected in 15 mL Falcon tubes and treated in the same manner as the ones for
DO isotope measurements, except for preservation with $HgCl_2$. All samples were stored in a mobile refrigerator
box at 4 °C directly after collection and carried to the laboratory where they were measured within 24 h.

**2.3 Identification of possible mineral precipitates**

In order to determine possible mineral precipitate data for the pH, pe (activitythe negative decadic activity of
available electrons), temperature, alkalinity (as $CaCO_3$), as well as cations and anions, the specific sampling
points were fed into the program PhreeqC (Version 3; Parkhurst and Appelo, 2013) for calculation of saturation
indices. The database used for these calculations was Wateq4.

**2.5 Laboratory methods**

**2.5.1 Identification of cyanobacteria**

Samples were collected in a preliminary field assessment at the anoxic piping where the spring flows into the creek
(E2), in the middle of the creek at the first small pond after the water had contact to the atmosphere (E3) and about
5 m downstream of this pond from an algal mat with bubbles on the surface (E4). Samples for cyanobacterial
isolation were collected in sterile 2-mL-Sarsted tubes and sealed. Samples for microscopic analysis were collected
with a 75 % ethanol sterilised spatula and placed in a sterile 6 cm petri dish (Sarsted, Germany). Immediately after
returning from sampling, samples were embedded in 1.5 % Agarose in de-ionized water to preserve the structure
of the bio mats during further handling and shipping.
Microscopic analysis was performed on thin sections of the embedded mats using a CLSM-type microscope (LSM
880, Carl Zeiss), using modified acquisition settings from Jung *et al.* (2019) to discriminate between cyanobacterial
(chlorophyll-*a* (chl-*a*) and phycobiliproteins (PBP) and green algal (chl *a*) fluorescence. Laser transmission images
were also generated using the 543 nm laser.
A spatula tip of green coloured mat was used to inoculate 5 mL of BG11 medium (Stanier et al., 1971) in a well
of a 6-well plate and incubated for 3 weeks at 24 °C on a 16:8 day:night cycle with illumination at 15 µmols
photons $m^2/s$ under an OSRAM L30W/840 LUMINLUX Cool White bulb. Individual Cyanobacterial species were
picked from the mat cultures under a Nikon SMZ-U Zoom binocular microscope for further subculturing on 1 %
agar solidified BG11 plates, as well as liquid culture. Isolates were observed under an Olympus BX53 light
microscope and their morphologies recorded using an Olympus DP26 Camera. The number of cells per filament
and cell dimensions were measured using ImageJ 1.47v software. DNA was extracted (Gehringer et. al, 2010)
from one axenic isolate of a microscopically identified *Persinema* species of cyanobacteria. The 16s rDNA gene
and intergenic spacer sequence was amplified by the SSU-4 fwd and ptLSU-C-D rev primer pair (Marin et al.,
2005) using the Taq PCR mastermix (Qiagen, Germany). The PCR product was purified (NucleoSpin PCR clean-
up kit, Macherey-Nagel, Germany) and sequenced (Wilmotte et al., 1993). Sequences were merged (HVDR
Fragment Merger tool, Bell & Kramvis, 2013) and the final 16S-ITS sequence submitted to National Center for
Biotechnology Information, National Institute of Health, USA (NCBI).

**2.5.2 Isotope measurements**

Stable isotope ratios of DO (expressed as $\delta^{18}O_{DO}$) were measured on a Delta V Advantage Isotope Ratio Mass
Spectrometer (IRMS; Thermo Fisher Scientific, Bremen, Germany) coupled to an automated equilibration unit
(Gasbench II). Measurements were carried out in continuous flow mode with a modified method by Barth et al.
(2004). Here the isolation of DO into a headspace relies on a helium extraction technique by Kampbell et al.
(1989) and Wassenaar and Koehler (1999). Different portions of laboratory air were injected into helium-flushed
Exetainers and used to correct obtained data sets for linearity and instrumental drift during each run. Here
laboratory air is defined to represent atmospheric oxygen with a ubiquitous value of 23.9 ‰ versus Vienna
Standard Mean Ocean Water (VSMOW) (Barkan and Luz, 2005). Data were normalized to this value.

$\delta = (R_{sample}/ R_{SMOW} - 1)$ (Clark and Fritz, 1997)

To obtain ratio changes in per mil (‰), the $\delta$ values were multiplied by factor of 1000.
All samples were measured in triplicates and isotope values standard deviations (1σ) were less than 0.1 and 0.2 ‰
for $\delta^{18}O_{H2O}$ and $\delta^{18}O_{DO}$, respectively.
**3 Results and discussion**
**3.1 On-site parameters**
The on-site parameters as displayed in Table 1 show a range of pH values between 6.1 and 8.6 in the cold season

| Sampling point | Distance from spring (m) | pH | O$_2$ (mg/L) | Temperature (°C) | Conductivity (mS/cm) | Alkalinity (mg/L) | Fe$^{2+}$ (mg/L) | Fe$^{3+}$ (mg/L) | Na$^+$ (g/L) | Ca$^{2+}$ (g/L) | SO$_4^{2-}$ (g/L) | Cl$^-$ (g/L) | U$^{6+}$ (µg/L) |
|---|---|---|---|---|---|---|---|---|---|---|---|---|---|
| **Cold season** | | | | | | | | | | | | | |
| E1a | 0 | 6.1 | 2.3 | 19.5 | 16.8 | 820 | 6.6 | 0.4 | 2.5 | 1.2 | 2.1 | 4.4 | 170 |
| E1b | 0 | 6.5 | 3.4 | 19.3 | 16.4 | 828 | 6.6 | 0.4 | 2.5 | 1.2 | 2.2 | 4.5 | 170 |
| E2 | 15 | 6.5 | 4.5 | 19.3 | 16.5 | 796 | 5.6 | 0.4 | 2.5 | 1.2 | 2.2 | 4.5 | 170 |
| E3 | 45 | 6.7 | 5.8 | 17.5 | 16.8 | 790 | 5.7 | 0.4 | 2.5 | 1.2 | 2.2 | 4.5 | 170 |
| E3.1 | 65 | 6.5 | 7.4 | 17.3 | 16.6 | 810 | 4.5 | 0.5 | 2.5 | 1.2 | 2.2 | 4.5 | 170 |
| E4 | 85 | 7.1 | 8.0 | 16.2 | 16.9 | 804 | 3.9 | 0.6 | 2.5 | 1.2 | 2.2 | 4.5 | 170 |
| E4.1 | 115 | 7.5 | 8.7 | 16.1 | 17.0 | 808 | 3.4 | 0.6 | 2.5 | 1.2 | 2.2 | 4.5 | 170 |
| E5 | 145 | 7.9 | 8.9 | 15.2 | 16.8 | 804 | 0.9 | 0.8 | 2.4 | 1.2 | 2.2 | 4.5 | 170 |
| E5.1 | 175 | 7.6 | 9.1 | 15.3 | 16.8 | 816 | 0.4 | 0.5 | 2.5 | 1.2 | 2.2 | 4.5 | 170 |
| E6 | 205 | 7.9 | 9.5 | 14.1 | 16.9 | 760 | 0.2 | 0.1 | 2.5 | 1.2 | 2.2 | 4.5 | 170 |
| E6.1 | 235 | 7.9 | 9.7 | 13.3 | 16.5 | 770 | 0.0 | 0.1 | 2.5 | 1.2 | 2.2 | 4.5 | 170 |
| E7 | 265 | 8.0 | 10.1 | 12.3 | 16.6 | 760 | 0.0 | 0.1 | 2.4 | 1.1 | 2.2 | 4.5 | 170 |
| E8 | 295 | 8.0 | 10.5 | 10.8 | 1.1 | 195 | 0.0 | 0.1 | 0.1 | 0.1 | 0.1 | 0.2 | 5.0 |
| E9 | 300 | 8.6 | 11.0 | 7.4 | 0.5 | 160 | 0.0 | 0.1 | 0.0 | 0.1 | 0.0 | 0.0 | 0.4 |
| **Warm season** | | | | | | | | | | | | | |
| E1a | 0 | 6.3 | 3.6 | 21.3 | 16.3 | 874 | 6.9 | 0.0 | 2.4 | 1.1 | 2.2 | 4.5 | 190 |
| E1b | 0 | 6.4 | 3.9 | 21.2 | 16.4 | 850 | 6.7 | 0.1 | 2.5 | 1.1 | 2.2 | 4.4 | 190 |
| E2 | 15 | 6.5 | 5.9 | 20.6 | 16.4 | 846 | 5.6 | 0.0 | 2.5 | 1.1 | 2.1 | 4.4 | 160 |
| E3 | 45 | 6.6 | 6.6 | 21.6 | 16.4 | 814 | 4.0 | 0.0 | 2.4 | 1.1 | 2.2 | 4.4 | 160 |
| E3.1 | 65 | 6.9 | 7.6 | 22.5 | 16.4 | 808 | 2.9 | 0.1 | 2.4 | 1.1 | 2.2 | 4.4 | 160 |
| E4 | 85 | 7.2 | 8.2 | 22.7 | 16.4 | 826 | 1.5 | 0.1 | 2.5 | 1.1 | 2.2 | 4.4 | 160 |
| E4.1 | 115 | 7.3 | 8.0 | 23.0 | 16.4 | 812 | 0.7 | 0.2 | 2.5 | 1.1 | 2.2 | 4.5 | 160 |
| E5 | 145 | 7.4 | 8.1 | 24.0 | 16.4 | 786 | 0.1 | 0.1 | 2.4 | 1.1 | 2.2 | 4.5 | 160 |
| E5.1 | 175 | 7.5 | 8.0 | 25.6 | 16.4 | 804 | 0.0 | 0.0 | 2.5 | 1.1 | 2.2 | 4.5 | 160 |
| E6 | 205 | 7.5 | 8.1 | 25.7 | 16.4 | 796 | 0.0 | 0.0 | 2.5 | 1.1 | 2.2 | 4.5 | 160 |
| E6.1 | 235 | 7.5 | 7.9 | 25.5 | 16.4 | 748 | 0.0 | 0.0 | 2.4 | 1.1 | 2.2 | 4.5 | 150 |
| E7 | 265 | 7.5 | 8.1 | 24.9 | 16.4 | 742 | 0.0 | 0.0 | 2.5 | 1.1 | 2.2 | 4.5 | 150 |
| E8 | 295 | 7.5 | 8.3 | 22.8 | 16.4 | 708 | 0.0 | 0.0 | 2.5 | 1.1 | 2.2 | 0.3 | 180 |
| E9 | 300 | 8.0 | 8.8 | 16.8 | 0.8 | 238 | 0.0 | 0.0 | 0.1 | 0.1 | 0.0 | 0.0 | 1.0 |

**Table 1** On-site parameters: major ion concentrations and Fe(II) and DO concentrations for the Espan Spring. Note that values before the forward slash are for cold season and after the slash for warm season.

and between 6.3 and 8.0 in the warm season. The observed changes of the pH over the course of the spring are mostly due to the constant degassing of $CO_2$ from the spring. Oxygen values range from 2.3 mg/L to 11.0 mg/L in the cold season and from 3.6 mg/L to 8.8 mg/L in the warm season. Differences between the cold and warm season are due to the fact that cold water can dissolve more $O_2$ than warm water. The general increase in the amount of DO over the course of the spring is due to a continuous dissolution of atmospheric $O_2$ in the spring water and due to the impact of photosynthesis. Water temperatures ranged between 19.3 and 7.4°C in the cold season and between 21.3 and 25.7°C in the warm season. The conductivity remained stable over the course of the spring and only showed minor differences between the cold and warm season. The same applies to the alkalinity. The behavior of the Fe(II) and Fe(III) is described in section 3.5. Values of major ions ($Cl^-$, $SO_4^{2-}$, $NO_3^-$, $Na^+$, $K^+$, $Ca^{2+}$ and $Mg^{2+}$) remained constant over the course of the spring and show no differences between the cold and warm season.

**3.2 Precipitation calculations**

Precipitating mineral phases as determined with PhreeqC showed that the dominant phase at all measurement points was Hematite ($Fe_2O_3$) (Supplementary Information Table S1 and S2). Additionally, Goethite (α-$\underline{FeO(OH)}$), Ferrihydrite ($Fe(OH)_3$), Siderite ($FeCO_3$) and K-Jarosite ($KFe^{3+}_3(OH)_6(SO_4)_2$) as well as $CaCO_3$ and Rhodochrosite ($MnCO_3$) showed elevated SI values and indicated precipitation.

**3.3 Bacterial contents**

Confocal Laser scanning microscopy (CLSM) showed that only the samples from Site E4.1 have photosynthetic organisms in significant quantities during the cold period. The photosynthetic community in this biofilm was dominated by cyanobacteria, with very few eukaryotic algae (Fig. 2). Lyngby was observed along the sides of the fast-flowing stream on the smooth hard canal section at E2, however, the loosely built *Lyngbya* sp. mats were only observed in the wider, shallower sections from sampling sites E3 to E5, and predominating between sites E3.1 and E4.1. The *Lyngbya* sp. filaments were not encrusted by oxidized iron as proven by light microscopy. As these are simple cyanobacterial mats on top of loose iron oxides, with no additional microbial layers beneath them, the bubbles are presumably oxygen generated during photosynthesis (Supplementary Information Fig. 1)

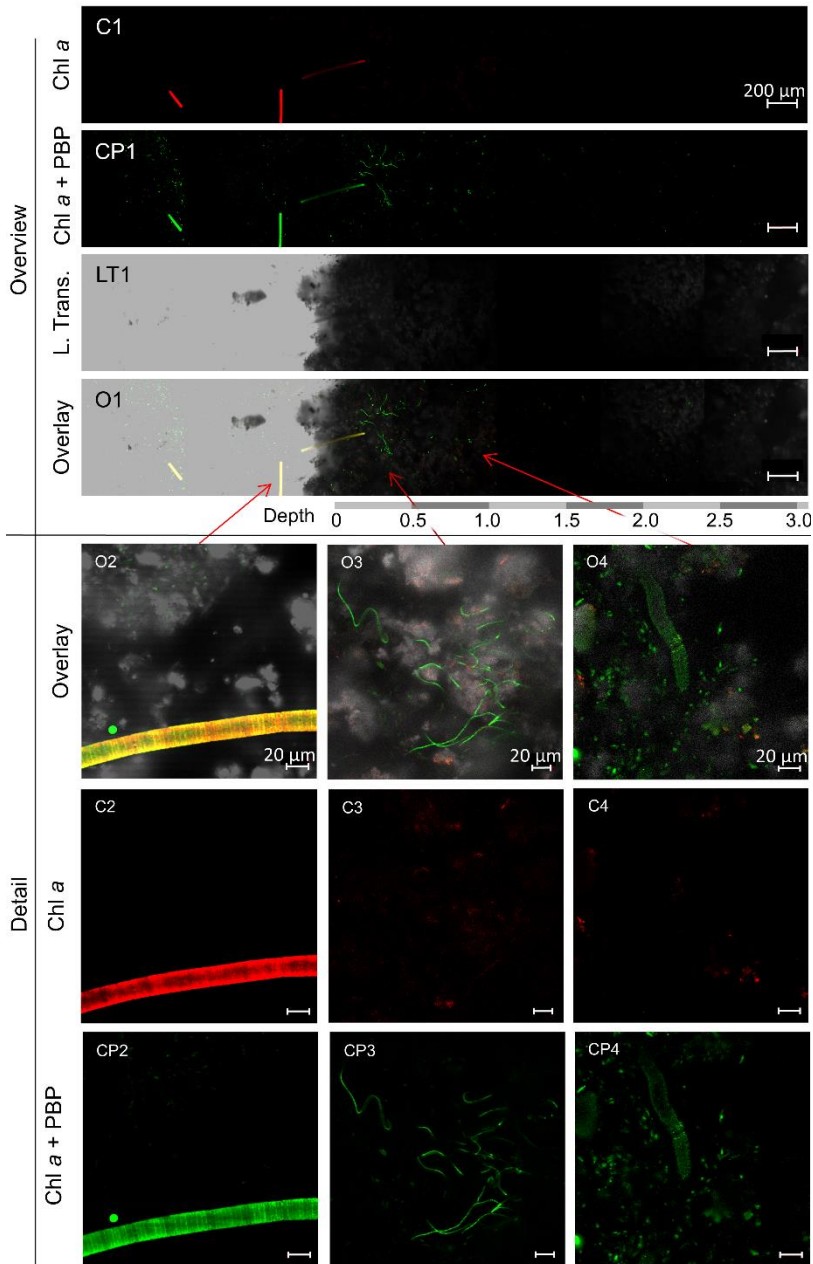

239

**Figure 2** CLSM images of mat sample E4.1. Overview**:** images of the cross-section of the top 3 mm of the biofilm with the chl-*a* (C1) and chl-*a* plus PBP (CP1) fluorescence profile, complemented by a laser transmission picture (LT1) and the superimposed image (O1). **Detail:** Superimposed images (O2/3/4) of chl-*a* (C2/3/3) and chl-*a* plus PBP (CP2/3/4) fluorescence and laser transmission (not shown) of distinct organisms found in the bio mat. **O2**: eukaryotic algae. **O3**: Possible Klisinema- or Persinema–like sp. and a unicellular cyanobacterium. **O4**: Lynbya – like sp. and a unicellular cyanobacterium.

Most of the cyanobacteria and all eukaryotic algae were located in the topmost 1.2 mm of the biofilms (Fig. 2 O1). Close-up images show eukaryotic algae (Fig. 2.O2), thin filamentous cyanobacteria, possibly *Persinema sp.* or *Klisinema sp.* (Fig. 2.O3) and *Lynbya* sp. (Fig. 2.O4). All pictures of the top layers of this sample site show an abundance of unidentified unicellular cyanobacteria, while images from the other sample sites show very few photosynthetic organisms at all (supplementary information Fig. 2).

In order to determine the identity of the predominant cyanobacterial species isolated from the E4.1 enrichment cultures, a determination key was used to compare particular features of an isolate to those already in the literature for specific cyanobacterial species (Komárek und Anagnostidis, 2005). Note that enrichment cultures for samples E2 and E3 did not yield enough material for cyanobacterial determination after 5 weeks in culture.

The red-brown filamentous strain (Fig. 3, c, d) exhibits single filaments, without false branching, that are 30.9 to 38.2 µm wide (Table 2), with a firm, 9.5 to 14 µm thick sheath. The trichomes and single cells are 21.5 to 24.2 µm wide and 1.5 to 4.1 µm long (Table 2), are red-brown in colour and constricted at the cross-walls. Based on these characteristics, the species was attributed to the cyanobacterial genus *Lyngbya*.

| | Filament length | Filament width (µm) | Cell width (µm) | Cell length (µm) |
|---|---|---|---|---|
| *Lyngbya sp.* | Indeterminate | 30.9 – 38.2 | 21.5 – 24.2 | 1.5 – 4.1 |
| *Klisinema sp.* | Indeterminate | 3.9 – 7.6 | 12 – 4.5 | 0.3 – 0.4 |
| *Persinema. sp* | Indeterminate | | 0.5 – 1.8 | 2.7 – 4.7 |

**Table 2** Filament and cell dimensions of the proposed cyanobacterial species.

The blue-green filamentous strain (Fig. 3, b) produces single filaments, without false branching, that are 3.9 to 7.6 µm wide (Table 2) with a firm, 2.7 to 3.1 µm thick sheath. The trichomes and single cells are 1.2 to 4.5 µm wide and 0.3 to 0.4 µm long (Table 2), blue-green in colour without constriction at the cross-walls. The terminal cells in mature filaments are conical, elongated and bent to one side, corresponding to those of the *Klisinema* genus recently described by Heidari et al. (2018). The thin, naked pale green filaments (Figure 3a & e) resembled those of *Persinema komarekii* (Heidari et al., 2018) with apical cells flattened at the end. In contrast to the observations of Heidari et al. (2018), we observed terminal aerotopes. This species was purified in culture and the 16S-ITS (NCBI accession number: MT708471) sequence confirmed its identity to *Persinema komarekii* (MF348313).

269

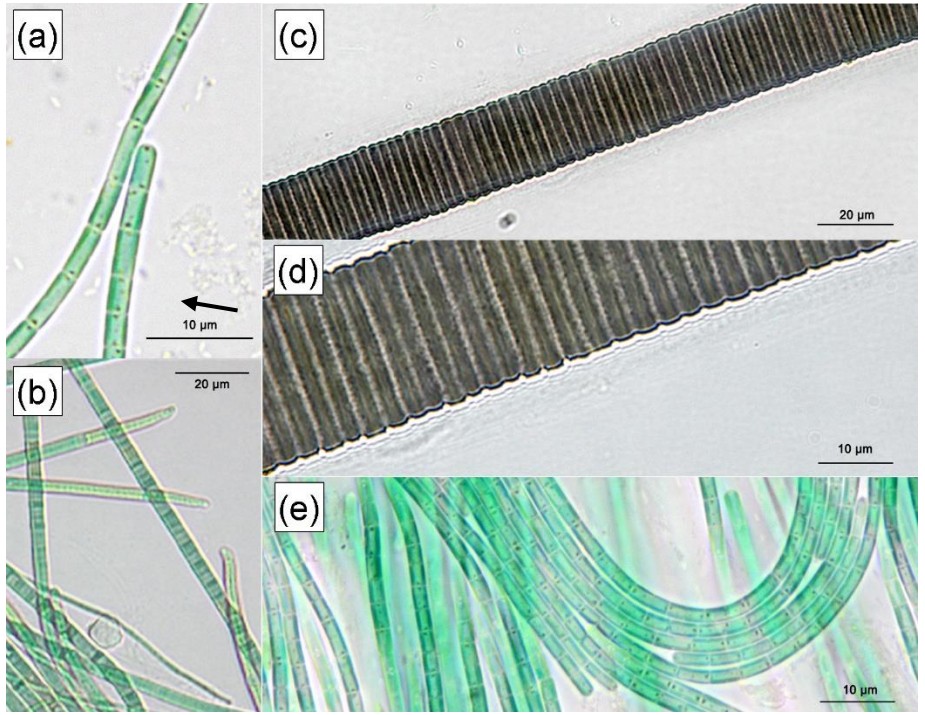

**Figure 3** Light Micrographs of the predominant isolates from sample E4.1: a) Single filament of *Persinema sp.,* arrow indicates aerotopes. b) Biofilm of *Klisinema sp.* interspersed with *Persinema sp* (arrow) c) *Lyngbya sp.*, filament d) *Lyngbya sp* sheath detail, E: Biofilm of *Persinema sp.*

**3.4 Dissolved oxygen (DO)**

The DO concentration in the Espan System was lowest at the faucet in the Pavilion (sampling point E1a) with a saturation of 25.3 % (2.3 mg/L) (Fig. 4a). Over the following 100 meters DO saturation increased to 88.1 % (8.7 mg/L) in sampling point E4.1. Afterwards the saturation continually increased to 94.6 % (11.0 mg/L) in point E8. From an initial depth of 435 meters with the abundance of reduced species such as Fe(II) and Mn(II), the low DO content in sampling point E1a was expected and in the further course, more atmospheric oxygen was able to dissolve. In addition, gas bubbles were observed in association with the *Lyngbya* mats. They were most prominent at sample site E4.1 and indicate a significant contribution of $O_2$ from daytime photosynthesis. However, saturation with DO was not reached during either of the sampling campaigns.

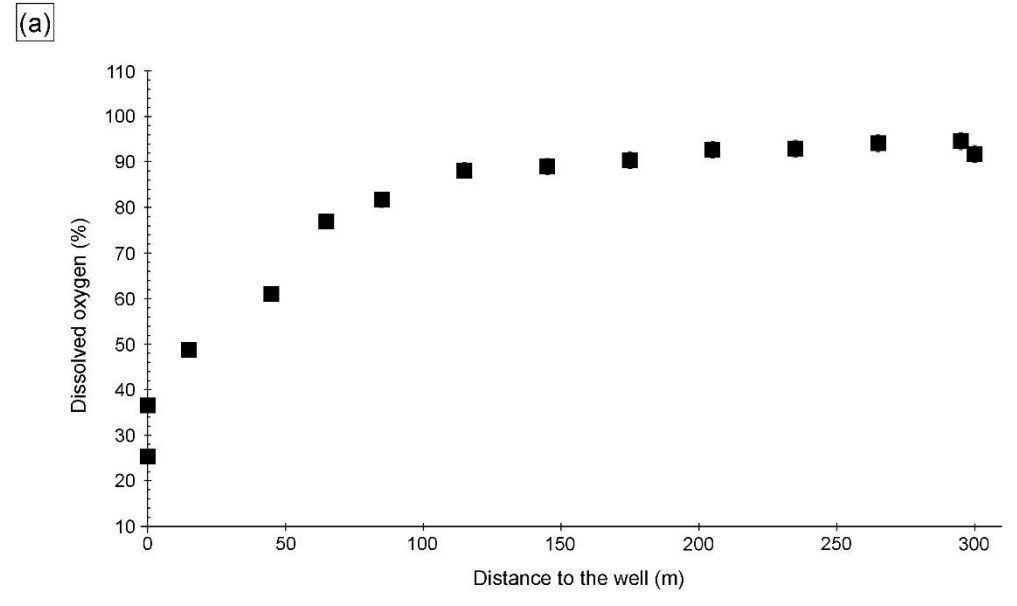

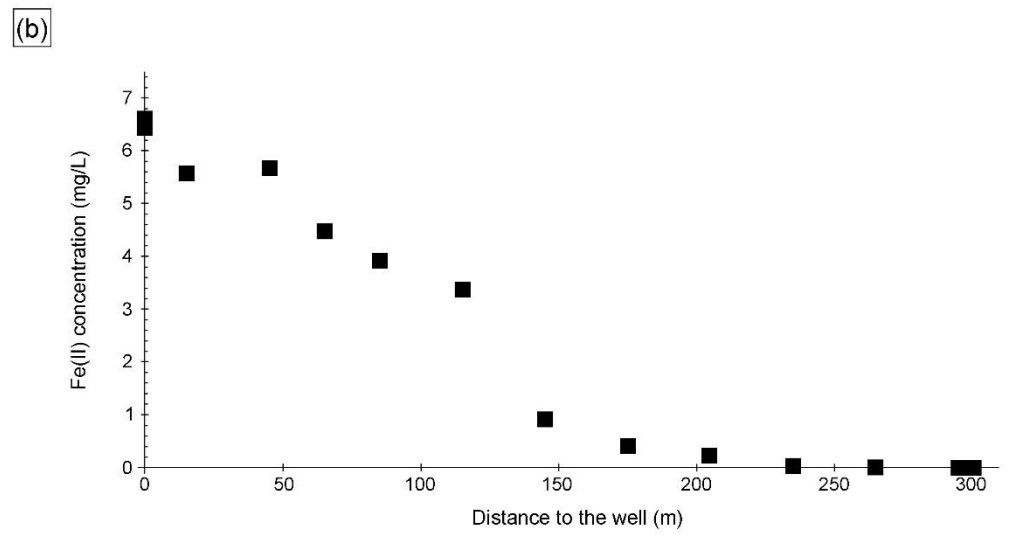

**Figure 4** a) Dissolved oxygen (%) and b) Fe(II) concentrations over the course of the Espan System in an example
graph for the cold season In February. The error for DO was 2 % and for Fe(II) it was 0.06 mg/L. Errors are within
symbol size.

**3.5 Fe(II) and Fe(III)**

The Fe(II) content was highest at the faucet with 6.6 mg/L while its lowest content was below instrument precision.
at sampling point E9 at 300 meters from the source (Fig. 4b). Fe(II) concentrations decreased constantly over the
stream course and were accompanied by increases in DO saturation (Fig. 4a). The decrease in Fe(II) could have
been caused by three major processes:

(1) Oxidation of Fe(II) to form ferric iron minerals such as ferrihydrite, hematite and goethite

(2) Precipitation of Fe(II) minerals such as the iron carbonate siderite ($FeCO_3$) and/or an amorphous ferrous silicate
phase or

(3) Adsorption of Fe(II) on already formed iron minerals.

All three possibilities seem plausible when taking into consideration the saturation indices of ferric iron minerals, goethite, ferrihydrite and hematite precipitate at all sampling points in the system (Köhler et al. 2020). These calculations furthermore show that siderite can precipitate in almost all sampling points while iron-silicate minerals are unlikely to precipitate. Therefore, adsorption of Fe (II) onto minerals is also a possible mechanism in the Espan System. Such adsorption of Fe(II) onto (oxyhydr)oxides was shown to typically occur under neutral conditions and should increase with rising pH (Zhang et al.,1992; Liger et al., 1999; Appelo et al., 2002; Silvester et al., 2005). Moreover, of large amounts of sulphate and chloride with average values of 2.2 and 4.5 g/L may have been responsible for maintaining observed high dissolved Fe(II) contents of the spring system at circum-neutral pH despite rising DO concentrations. Such elevated $Cl^-$ and $SO_4^{2-}$ contents can delay abiotic Fe(II) oxidation (Millero, 1985).

Dissolved Fe(III) was highest (0.8 mg/L) at sampling point E5 after 145 m and lowest (0.05 mg/L) at sampling point E7 after 265 m flow distance from the spring. The values initially increased from 0.4 mg/L in E1a to a maximum of 0.8 mg/L in point E5 (+/- 0.03 mg/L) and then decreased to their lowest value in sampling point E7. The solubility of iron oxides in natural systems at a circum-neutral pH and under aerobic conditions is generally very low (Cornell and Schwertmann, 2003) with values of the solubility product ($K_{sp}$) between $10^{-37}$ and $10^{-44}$ (Schwertmann, 1991). However, Fe(III) could still be detected in the water, thus showing that its dissolution was possible. The dissolution of iron oxides can occur through several pathways including as protonation, reduction and complexation that create Fe(III) cations, Fe(II) cations as well as Fe(II) and Fe(III) complexes (Schwertmann, 1991; Cornell and Schwertmann, 2003). Both the protonation as well as the reduction would lead to the formation of dissolved Fe(II). A steep increase in dissolved Fe(III) at 145 m downstream of the spring (from 0.5 mg/L to 0.8 mg/L) also indicated acceleration of this process. One reason for this increase could be available organic matter. However, further analyses are needed to verify this interpretation.

**3.6 $\delta^{18}O_{DO}$**

Figure 5 a) and b) show $\delta^{18}O_{DO}$ values over the course of the spring for the cold and warm seasons, respectively. The curves are divided into two zones for the cold season and three zones for the warm season.

*Zone 1*

In the cold season, zone 1 extended from sampling point E1a to point E4. In these first 85 meters, the $\delta^{18}O_{DO}$ increased from a value of +23.7 ‰ at the faucet (E1a) to + 25.7 ‰ at E4. In the warm season, zone 1 extended from E1a to E2 with only 15 m distance from the spring. In this zone the values increased from + 23.4 ‰ at the faucet to a maximum value of + 24.7 ‰ at E2. In both seasons, $\delta^{18}O_{DO}$ values at E1a were below the value expected for atmospheric equilibration (+ 24.6 ‰). At first sight such $^{16}O$-enriched $\delta^{18}O_{DO}$ values would suggest photosynthetic input of DO. However, the water originated from greater depths without any exposure to light and thus any photosynthetic influence can be ruled out.

The occurrence of $\delta^{18}O_{DO}$ values below + 24.6 ‰ in groundwater has been described in the literature (Wassenaar and Hendry, 2007; Smith et al., 2011; Parker et al., 2014 and Mader et al, 2018) and several explanations for this phenomenon have been suggested (Wassenaar and Hendry, 2007; Smith et al., 2011; Parker et al., 2014 and Mader et al, 2018). These include:

(1) possible transfer of photosynthetic or diffusive oxygen into the shallow aquifer (Smith et. al; 2011; Parker et
al., 2014; Mader et al., 2018),
(2) radial oxygen loss of plant roots (Teal and Kanwisher, 1966; Michaud and Richardson, 1989; Caetano and
Vale, 2002; Armstrong and Armstrong, 2005b)
(3) radiolysis of water (Wassenaar and Hendry, 2007) and
(4) kinetic gas transfer (Benson and Krause, 1980; Knox et al. ,1992; Mader et al. 2017)
Explanations (1) and (2) are very unlikely in the Espan Spring, because the water originates from a depth of 435
meters below ground through pipes that presumably prevent any exchange with surface water or possible impacts
of plant roots. It should however be noted that water from the Espan Spring contains up to 170 µg/L of uranium
from easily soluble uranium compounds that are commonly encountered in the Buntsandstein formations (Büttner
et al. 2006; Meurer and Banning, 2019). The geogenic radiation in the area is rather high because of the high
uranium content in the Variscian bedrocks of the area (Schwab, 1987; Büttner et al., 2006). Because of this,
radiolysis could be a possible explanation for the unexpected low $\delta^{18}O_{DO}$ values. Kinetic gas transfer of
atmospheric oxygen during transport in the pipes or at the faucet might another explanation, since the sample in
E1a is strongly DO undersaturated. During non-equilibrium gas exchange the kinetically faster $^{16}O$ would cause
$\delta^{18}O_{DO}$ below +24.6 ‰ until equilibrium is established (Benson and Krause, 1980; Knox et al. ,1992, Mader et al.

351 2017).


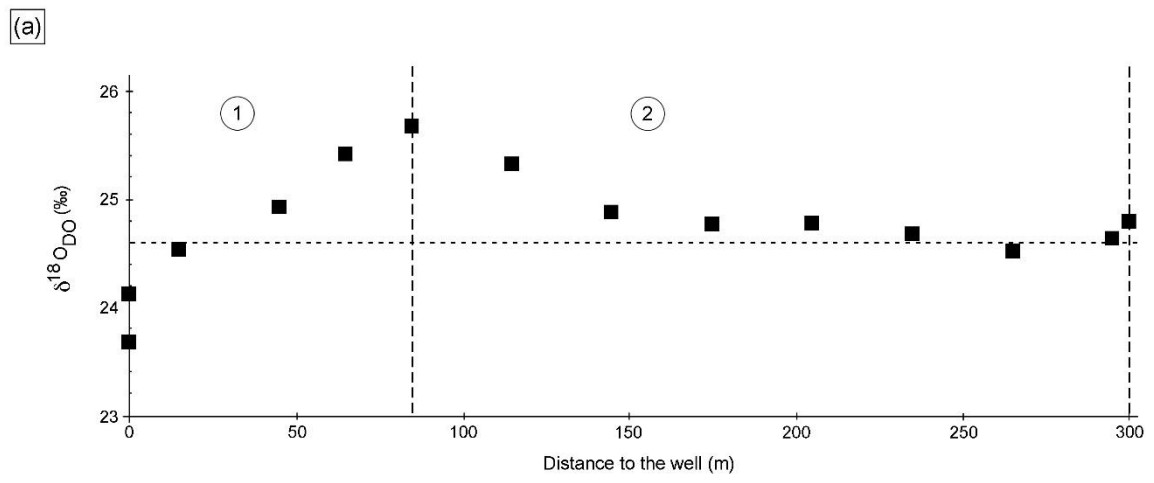

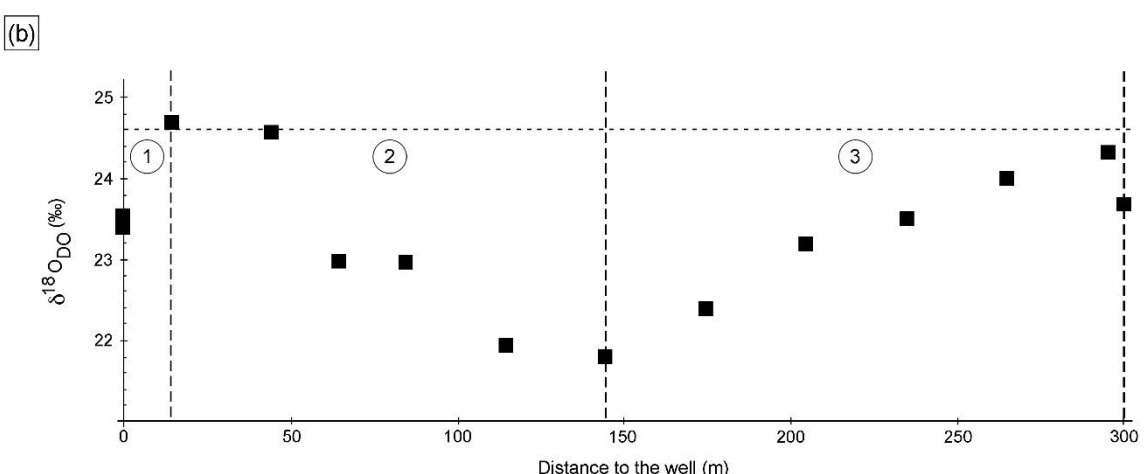


**Figure 5** $\delta^{18}O_{DO}$ in the cold season a) and the warm season b) over the course of the Espan Spring and stream
system with the atmospheric equilibrium value of + 24.6 ‰ marked by the horizontal line. Dashed vertical lines
show borders of the different zones of the fields labelled with 1, 2 and 3. The symbol size is larger than the error
bars.
Increases in $\delta^{18}O_{DO}$ values in zone 1 were accompanied by increases in DO (Fig. 6a). In the cold season, a strong
positive correlation was evident between points E1a and E4. However, in the warm season, the same correlation
could only be observed between points E1a and E2 (Fig. 6b). Equilibration with the atmosphere would be
reasonable explanation for this trend until atmospheric equilibration was reached between point E2 and E3.
However, the $\delta^{18}O_{DO}$ values, at least in the cold season, increased above this threshold to a value of + 25.7 ‰.
This shows that another process in addition to atmospheric equilibration must have influenced the $\delta^{18}O_{DO}$ values
in zone 1. In the warm season, this was less evident, and the isotope atmospheric equilibrium value was only
marginally exceeded and remained within the range of the analytical uncertainties.

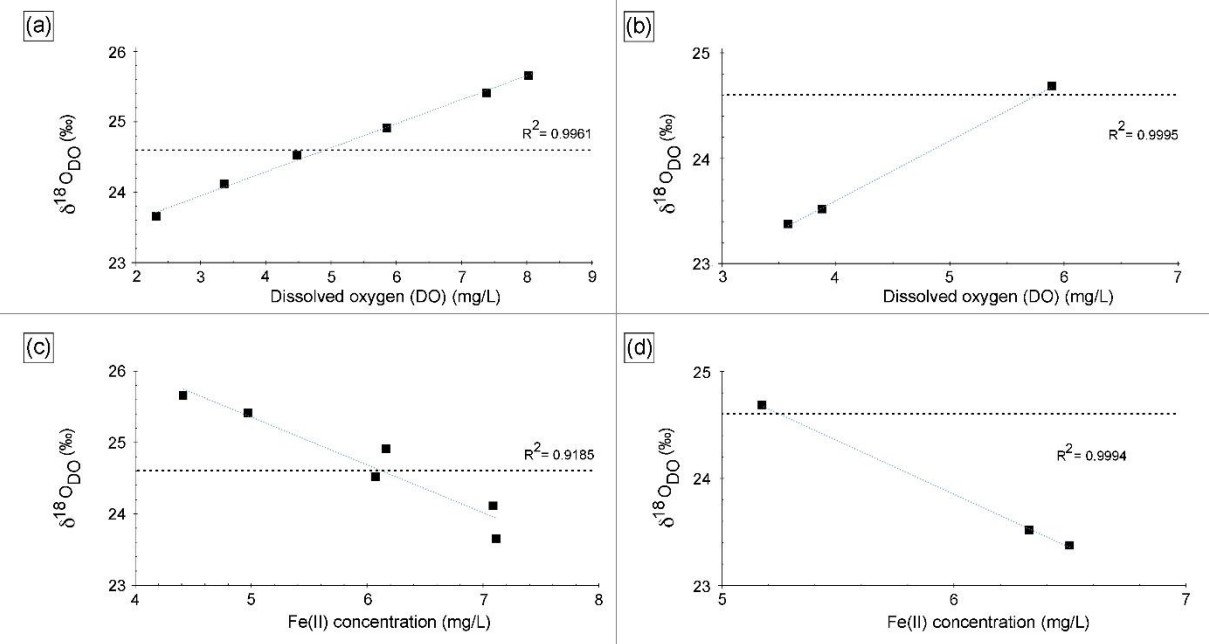


**Figure 6** Correlation between $\delta^{18}O_{DO}$ and DO over the course of the spring for zone 1 in the cold season a) and
the warm season b). Correlation between $\delta^{18}O_{DO}$ and Fe(II) contents over the course of the stream for zone 1 in
the cold season c) and the warm season d).

Even though these processes consume DO, both respiration and iron oxidation could be responsible for this trend
when assuming that they influence the $\delta^{18}O_{DO}$ values, while DO concentrations are constantly replenished by the
atmosphere. A direct negative correlation between Fe(II) concentrations and $\delta^{18}O_{DO}$ values between point E1a and
E4 was evident for cold season samples and in point E1a and E2 for warm season samples as shown in Figure 6c
and d. This correlation between Fe(II) and $\delta^{18}O_{DO}$ in the Espan System corresponds with the experimental
observations of Oba and Poulson (2009), as well as those of Pati et al. (2016). These studies demonstrate that Fe
oxidation leads to increases in $\delta^{18}O_{DO}$ values due to preferential consumption of $^{16}O$. The increase in $\delta^{18}O_{DO}$ due
to iron oxidation in a natural system, which is constantly supplied with fresh oxygen, indicates that Fe(II) oxidation
must be a dominant control on $\delta^{18}O_{DO}$ in the first 85 meters of the stream in the cold season and in the first 15
meters in warm season. It also implies that the direct impact of oxygen addition is subordinate in terms of DO
stable isotope changes. This is shown by  iron oxidation being the dominant factor that controls $\delta^{18}O_{DO}$ values,
even though oxygen is constantly supplied from the atmosphere.

*Zone 2*
In the cold season, zone 2 extended from sampling point E4 to point E9 with only minor variations in $\delta^{18}O_{DO}$. In
this zone, the $\delta^{18}O_{DO}$ decreased from + 25.7 ‰ in sampling point E4 to values around atmospheric equilibrium
with + 24.5 ‰ in E7 and + 24.8 ‰ in the Pegnitz River (Fig. 5a).
In the warm season, zone 2 extended from sampling point E2 to point E5 at 145 meters distance from the spring.
In this zone the values decreased from + 24.7 ‰ to a minimum value of + 21.8 ‰ in sampling point E5 (Fig. 5b).
This decrease in $\delta^{18}O_{DO}$ values can be explained by (1) a decrease of the impact of iron oxidation on the $\delta^{18}O_{DO}$
values and (2) a rising impact of atmospheric or photosynthetic oxygen. Even though a decrease in Fe(II) values
was still evident between E4 and E7 in the cold season, as well as between E2 and E5 in the warm season, it is
possible that the decrease was not caused by Fe(II) oxidation and subsequent precipitation as iron oxides.
Alternatively, the decrease could have been caused by adsorption of dissolved Fe(II) onto already existing iron
oxides such as goethite, ferrihydrite and hematite (Zhang, et al., 1992; Liger et al., 1999; Appelo et al., 2002;
Silvester et al. 2005). Because adsorbed Fe(II) is very resistant to oxidation (Park and Dempsey, 2005) the impact
of iron oxidation on the $\delta^{18}O_{DO}$ values would have decrease.
No significant changes in the water chemistry were evident and it can be assumed that after sampling point E2
(warm season) or E4 (cold season), a critical value was exceeded with enough Fe(II) having been adsorbed onto
iron oxides. In this case, iron oxidation --- while probably still taking place at small rates--- is no longer an
important factor dominating the $\delta^{18}O_{DO}$ values. Downstream of point E2 and E4 oxygen addition by the atmosphere
or by photosynthesis would become more important.
Intensive growth of cyanobacterial and algal mats were observed between point E3.1 and E5 in the cold season
and between E3 and E5 in the warm season (Fig. 1f). Because of this growth it can be postulated that in addition
to the atmospheric $O_2$ input, the $\delta^{18}O_{DO}$ values were also influenced by photosynthetically produced oxygen. While
this effect should be less pronounced in the cold and darker season, a stronger influence of photosynthetic oxygen
on the $\delta^{18}O_{DO}$ values would be expected in the warm season with higher light intensity. Such growth of
photosynthetic organisms in the Espan System is not surprising with iron being an important micronutrient
(Andrews et al., 2003).
The fact that photosynthesising organisms seem to preferentially grow and impact the $\delta^{18}O_{DO}$ values between
sampling point E3 and E5 may be due to the availability of Fe(II). In addition, the growth could also be controlled
by changes in the pH or other environmental influences, with the site being located in a public park with the
associated perturbations. Cyanobacteria, especially aquatic strains prefer a neutral to alkaline pH (Brock, 1973)
and the shift to higher pH values in this zone could be one of the main factors that drive increased supply of
cyanobacterial $O_2$. For instance, *Lyngbya* spp. are diazotrophic cyanobacteria, capable of fixing nitrogen during
low availability of light, when local oxygen levels are low (Stal, 2012, p. 102). This Oxygen released through
oxygenic photosynthesis would immediately react with Fe(II) and lower the partial pressure of oxygen around the
organisms in a slow flowing stream. This could also favor biological nitrogen fixation and limit carbon loss by
reducing photorespiration. Additionally, the reduced oxygen partial pressure induced by Fe(II) oxidation may
minimize the oxygenase activity of ribulose 1,5-biphosphate carboxylase/oxygenase (Rubisco), thereby favoring
$CO_2$-fixation (Stal, 2012, p. 113).

A screening of microbial ecology in several iron-rich circum-neutral springs and experiments with the
cyanobacterium *Synechococcus* PCC 7002 (Swanner et al. 2015a) revealed that many cyanobacteria show optimal
growth between 0.4 – 3.1 mg/L Fe(II) and that concentrations above 4.5 mg/L become growth-limiting. The iron
concentrations between point E3.1 and E5 in the cold season and E3 and E5 in the warm season are thus
approximately in the range of optimal cyanobacterial growth. In order to establish a clear correlation between the
iron concentration and the decrease in $\delta^{18}O_{DO}$ values, experiments would need to be carried out with the organisms
found in the Espan System. These have so far have not been assessed for their behaviour under variable iron
concentrations.
*Zone 3*
In the warm season, zone 3 extended from sampling point E5 to point E8. In this zone the $\delta^{18}O$ values rose again
from + 21.8 ‰ to + 24.3 ‰ (Fig. 5B). The renewed increase in values can be explained by the influence of iron
oxidation, respiration and a decrease in photosynthetic activity. Because Fe-contents only decreased marginally,
it can be assumed that decreases in photosynthetic activities are responsible for increase in the $\delta^{18}O$ values. This
matches our observations that downstream of point E5, only little or no photosynthetic growth took place. Oxygen
that would dissolve in the water after point E5 would thus most likely stem from the atmosphere. This would also
explain the approach to the equilibrium value of + 24.6 ‰. Reasons for the observed decrease in cyanobacteria
are however not clear and may include changes in temperature, light intensity and shifts in nutrient availability.
The temperature did not change significantly in this part of the watercourse and is therefore unlikely to have caused
a decrease in photosynthetic oxygen production. In contrast, reduced light exposure could have been responsible
as downstream of point E5 trees shade the water course. A decrease in nutrient availability is difficult to determine
because nitrate and phosphate were below the detection limit in the entire spring. Iron starvation could also be a
possible reason for the decrease in activity because only ~0.005 mg/L Fe(II) was left in the system in the lowest
course of the stream.
**4 Conclusions**
Our study is the first systematic analysis of $\delta^{18}O_{DO}$ values as a function of iron contents and oxygenic
photosynthetic biofilms in a natural iron-rich spring. We were able to confirm from field samples that Fe-oxidation
leads to increases in $\delta^{18}O_{DO}$ values even though oxygen was constantly replenished by atmospheric input. As soon
as photosynthetic oxygen is produced in the system, the effect of iron oxidation on the $\delta^{18}O_{DO}$ values becomes
negligible and can no longer be detected. The fact that photosynthesis has a strong impact on the $\delta^{18}O_{DO}$ values in
specific areas of the system may be controlled by high Fe contents of the system. Similar iron-rich springs show
optimal growth rates of cyanobacteria in the range of 0.4 – 3.1 mg/L Fe(II). The presented $\delta^{18}O_{DO}$ values showed
that photosynthetic activity is also strongest in the Espan System within this range of concentrations.
To what extent the changing Fe concentrations (Fe(II)/Fe(III)) influence the growth of cyanobacteria and algae
occurring in the Espan System, requires further investigation. This would ideally include isolating the organisms
from the water course and studying them under varying experimental levels of Fe, pH and temperature while
monitoring the $\delta^{18}O_{DO}$ of the system. Further field studies with organic material from the stream bed in
combination with stable carbon isotopes would be promising to narrow down processes for carbon and oxygen
budgets in this environment.
**5 Author contribution**
Inga Köhler, David Piatka and Johannes Barth carried out the sample collection and water analysis for on-site and
isotope data. Raul Martinez carried out the calculation of the saturation index. Michelle Gehringer, Achim
Herrmann and Arianna Gallo performed the analysis and interpretation of cyanobacteria and algae data. Inga
Köhler prepared the manuscript with contributions from all co-authors
**6 Acknowledgements**
Funding for this project was made available by the German Research Foundation (DFG) in the Project IsoDO (BA
2207/15-1) awarded to Johannes Barth and GE2558/3-1 & GE2558/4-1 awarded to Michelle Gehringer. We also
thank Christian Hanke, Marlene Dordoni and Marie Singer for help with sampling and analyses. The authors
declare that they have no conflict of interest.

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
