# Peer review of "How are oxygen budgets influenced by dissolved iron and"

_Biogeosciences, 2021_

## Author Response (AR1)

| Reviewer #1 | Answers |
|---|---|
| **Abstract.** The abstract presents many results found in the experiments, which can confuse the reader and be a little exhausting as an initial reading. My suggestion here is to present the most relevant results and conclusions, without several explanations and theories about what is possibly happening in the studied environment. For instance, the sentences "This trend existed..." (line 16) and "This may be due..." (line 18) could be deleted without prejudice to the information provided in the abstract. With such changes, I believe this section of the paper can deliver a plainer and concise message to the reader. | We adapted the abstract. Overall we think think that leaving in the most important results would make the work more quantitative. |
| **Introduction.** The Figure 1e and 1f should be clarified. My suggestion here is to change Fig.1e and Fig.1f by other images where the sampling points along with the margins or levees can be seen together, not images without spatial reference of the environment. In addition, it would be beneficial for the quality of the work to insert the scale in these photos, as well as a table with the depth of the water column (which can be inserted in section 2.2 "Samplings procedures"). The English here should be revised as well (line 82). | 1) We thank the reviewer for the suggestion to substitute Fig.1e and f. Unfortunately, additional images of the sampling sites are not available. In Fig. 1e however, the margin of the spring can be seen.

2) A scale has been inserted into the photos.

3) We added the water depth of the spring but did not insert an additional table as the water depth was always between 8-10 cm. |
| Line 85. The word "exceptional" should be deleted. | The word was deleted |
| Line 88. "as a problematic"...? The authors should complete this sentence to link it with the next phrase to make it clear and avoid the repetition of "This is" at the beginning of both sentences. What is problematic? Environmental issue? The outcome of something? | We thank the reviewer for this comment and changed the text accordingly. |
| Lines 91 to 93. This part of the text can be rewritten. The presentation of the phrases in this way is a little confusing. | The text was re-written in this section |
| **Methods.** Line 103. 1C - switch the "C" to a lowercase letter to be standardized with the rest of the denominations in the text. | The "C" was switched to lowercase letter. |
| Line 133. Suggestion: Samples for microscopic analysis were... | The text was changed. |

| | |
|---|---|
| Line 142. Does this day-night lighting cycle correspond to that found at the sampling site or does it correspond to a standardized methodology for these analyses? | We applied a standard method for growing cyanobacteria in a laboratory setting to maximize growth, while maintaining the day/night cycle. |
| Line 150. It seems that there is some missing information in the sentence "The cleaned PCR product...". | We added the missing information. |
| Line 158. Could the authors provide the definition of "laboratory air"? | We provided the definition of laboratory air to the text. |
| **Results and discussion.** Line 194. Suggestion: On-site parameters measured in the Espan Spring: major ion concentrations, Fe(II) and DO concentrations. | We changed the text accordingly. |
| Line 217. Could the authors provide the scales of the images contained in Figure 2 more clearly? It is hard to see. Maybe increase the font size and change the colour from red to white. | We edited the figure accordingly. |
| Line 228. I did not find these images in the supplementary material. Please insert them in the file. | We inserted the image to the supplementary material. |
| Line 233 and Line 243. Delete the "&" and replace it with a comma. | We deleted the "&" and replaced it with a comma. |
| Line 249. B) - switch the "B" to a lowercase letter to be standardized with the rest of the denominations in the text. | The "B" was switched to lowercase letter. |
| Line 294. It would really be interesting to analyse these data together with the local organic matter data (including stable carbon isotope analysis). | This is an interesting point. However we did not sample organic matter data of the Espan Spring in this study. This aspect could be reserved for future work and we brought this point up in the conclusions.. |
| Line 320. Could the authors include some relevant references about the uranium occurrence in the Buntsandstein formations? Perhaps this work can help in some way: Meurer, M., Banning, A. Uranmobilisierung im Helgoländer Buntsandstein – Auswirkungen auf die Brack- und Trinkwasserqualität. Grundwasser 24, 43–50 (2019). | The citation was included it into the manuscript. |
| Line 338. Here a brief discussion of these "other possible processes" would be interesting. | The other processes have been discussed in detail in the text now. |
| Line 340. The font size used in the graphics must be increased. | The font size was increased. |

| | |
|---|---|
| Line 354. Could the authors briefly explain with more details the sentence "It also implies that the direct impact of oxygen addition is subordinate in terms of DO stable isotope changes."? | We have explained the sentences in more detail now. |
| Line 371. Perhaps that sentence would be plainer if it were rewritten. | The sentence was deleted. |
| Line 385. Do the authors consider any scenario considering the (low?)-fluid shear dynamics of the Espan System? This can be one important (among many existing) environment stimulus for the bacteria adaptation in this peculiar ecosystem, especially taking into account the secondary metabolism of the cells and the possible physiological and chemical responses. | Possible effects of fluid shear dynamics on the bacteria have now been added in this section. |
| | |
| **Reviewer #2** | **Answers** |
| This is an interesting work, particularly for its novelty as a pilot study and the first of its kind in the study region. The presentation or results, discussion, and conclusion are generally well-supported and presented, and I would recommend this study for publication pending minor revisions. Some specific edits are outlined below: | Thank you for the encouraging comments. |
| Lines 36 - 38: This sentence seems a bit long, and can be divided into two separate sentences. | We have separated the sentence. |
| Lines 42-43: This passage should be referenced. | We have added some relevant references to this section. |
| Figure 1: It may be more clear to overlay the sampling sites on the traced graphical map, rather than the satellite image. | The figure was not changed as overlaying the sampling sites on the graphical map would have decreased the font sizes beyond readability. |
| Line 80: This should more correctly read "Overview of", rather than "Overview on" | We changed the wording here |
| Line 82: I assume this should read "image E shows" rather than "image is shows" | The wording was changed. |
| Line 87: Spelling error; "assess", rather than "asses" | We have corrected this mistake. |
| Line 99: A geological description of the Buntsandstein Formation would be useful here. | We have added a description of the Buntstandstein formation. |

| | |
|---|---|
| Line 125: The acronym "pe" should be defined here or elsewhere. | We have now explained the acronym. |
| Lines 131-132: Minor grammar corrections here. "...at first small pond..." should be corrected to "...at the first small pond...", and "...from algal mat..." should be "...from an algal mat...". | We have changed this accordingly. |
| Line 143: The word "bulb" here should not be pluralized as "bulbs". | We have changed this accordingly. |
| Lines 146 - 150: This sentence is too long, and can be split into two. | The sentence was split into two parts now. |
| Lines 150 - 151: This sentence appears to be a fragment and needs to be corrected. | The sentence was changed. |
| Line 164: "Per mille" should be written as "per mil". | Per mil was changed into the correct form. |
| Line 197: The passage "Oxygen values rise from..." should be corrected to "Oxygen values range from..." | The sentence was corrected. |
| Line 259: "Neither" should be replaced with "Either" to avoid a double-negative. | We exchanged neither for either. |
| Line 292: Minor grammar edit; "Steep increase" should be correct to "A steep increase". | The grammar was corrected. |
| Line 326: Minor grammar edit. "...until equilibrium establishes" should be corrected to "until equilibrium is established". | The grammar was corrected. |
| Line 385: What are some examples of these "associated perturbations"? | We explain the perturbations in the new text. |
| Line 413: Minor grammar edit. "Analyses" should be corrected to "analysis". | The grammar was corrected. |
| Line 417: Minor grammar edit. "cannot hardly be detected any longer" should be corrected to "can hardly be detected" or "can no longer be detected". | The grammar was corrected. |
| Lines 421 - 424: This sentence is too long, and can be split up into two sentences. | We shortened the sentence. |